# LEARNING FEATURES OF MUSIC FROM SCRATCH

**John Thickstun[1], Zaid Harchaoui[2] & Sham M. Kakade[1,2]**
[1] Department of Computer Science and Engineering, [2] Department of Statistics
University of Washington
Seattle, WA 98195, USA
{thickstn,sham}@cs.washington.edu, name@uw.edu

## ABSTRACT

This paper introduces a new large-scale music dataset, MusicNet, to serve as a source of supervision and evaluation of machine learning methods for music research. MusicNet consists of hundreds of freely-licensed classical music recordings by 10 composers, written for 11 instruments, together with instrument/note annotations resulting in over 1 million temporal labels on 34 hours of chamber music performances under various studio and microphone conditions.

The paper defines a multi-label classification task to predict notes in musical recordings, along with an evaluation protocol, and benchmarks several machine learning architectures for this task: i) learning from spectrogram features; ii) end-to-end learning with a neural net; iii) end-to-end learning with a convolutional neural net. These experiments show that end-to-end models trained for note prediction learn frequency selective filters as a low-level representation of audio.

## 1 INTRODUCTION

Music research has benefited recently from the effectiveness of machine learning methods on a wide range of problems from music recommendation (van den Oord et al., 2013; McFee & Lanckriet, 2011) to music generation (Hadjeres & Pachet, 2016); see also the recent demos of the Google Magenta project[1]. As of today, there is no large publicly available labeled dataset for the simple yet challenging task of note prediction for classical music. The MIREX MultiF0 Development Set (Benetos & Dixon, 2011) and the Bach10 dataset (Duan et al., 2011) together contain less than 7 minutes of labeled music. These datasets were designed for method evaluation, not for training supervised learning methods.

This situation stands in contrast to other application domains of machine learning. For instance, in computer vision large labeled datasets such as ImageNet (Russakovsky et al., 2015) are fruitfully used to train end-to-end learning architectures. Learned feature representations have outperformed traditional hand-crafted low-level visual features and lead to tremendous progress for image classification. In (Humphrey et al., 2012), Humphrey, Bello, and LeCun issued a call to action: "Deep architectures often require a large amount of labeled data for supervised training, a luxury music informatics has never really enjoyed. Given the proven success of supervised methods, MIR would likely benefit a good deal from a concentrated effort in the curation of sharable data in a sustainable manner."

This paper introduces a new large labeled dataset, MusicNet, which is publicly available[2] as a resource for learning feature representations of music. MusicNet is a corpus of aligned labels on freely-licensed classical music recordings, made possible by licensing initiatives of the European Archive, the Isabella Stewart Gardner Museum, Musopen, and various individual artists. The dataset consists of 34 hours of human-verified aligned recordings, containing a total of $1,299,329$ individual labels on segments of these recordings. Table 1 summarizes statistics of MusicNet.

The focus of this paper's experiments is to learn low-level features of music from raw audio data. In Sect. 4, we will construct a multi-label classification task to predict notes in musical recordings,

---

[1] https://magenta.tensorflow.org/
[2] http://homes.cs.washington.edu/~thickstn/musicnet.html.

**MusicNet**

| Minutes | Labels | Recordings | Error Rate |
|---|---|---|---|
| 2,048 | 1,299,329 | 330 | 4.0% |

| Ensemble | Minutes | Labels |
|---|---|---|
| Solo Piano | 917 | 576,471 |
| String Quartet | 405 | 259,702 |
| Accompanied Violin | 148 | 124,886 |
| Piano Quartet | 73 | 60,362 |
| Accompanied Cello | 63 | 37,557 |
| String Sextet | 48 | 33,248 |
| Piano Trio | 46 | 28,873 |
| Piano Quintet | 25 | 27,545 |
| Wind Quintet | 43 | 24,820 |
| Horn Piano Trio | 30 | 18,799 |
| Wind Octet | 23 | 14,635 |
| Clarinet-Cello-Piano Trio | 25 | 13,447 |
| Pairs Clarinet-Horn-Bassoon | 24 | 12,218 |
| Clarinet Quintet | 26 | 11,184 |
| Solo Cello | 49 | 10,876 |
| Accompanied Clarinet | 20 | 10,049 |
| Solo Violin | 30 | 8,837 |
| Violin and Harpsichord | 16 | 7,469 |
| Viola Quintet | 15 | 4,156 |
| Solo Flute | 8 | 2,214 |

| Composer | Minutes | Labels |
|---|---|---|
| Beethoven | 1,085 | 736,072 |
| Schubert | 253 | 146,648 |
| Brahms | 192 | 133,109 |
| Mozart | 156 | 99,641 |
| Bach | 184 | 62,782 |
| Dvorak | 56 | 46,261 |
| Cambini | 43 | 24,820 |
| Faure | 33 | 22,349 |
| Ravel | 27 | 21,243 |
| Haydn | 15 | 6,404 |

| Instrument | Minutes | Labels |
|---|---|---|
| Piano | 1346 | 794,532 |
| Violin | 874 | 230,484 |
| Viola | 621 | 99,407 |
| Cello | 800 | 99,132 |
| Clarinet | 173 | 24,426 |
| Bassoon | 102 | 14,954 |
| Horn | 132 | 11,468 |
| Oboe | 66 | 8,696 |
| Flute | 69 | 8,310 |
| Harpsichord | 16 | 4,914 |
| String Bass | 38 | 3,006 |

| | Piano | Violin | Cello | Viola | Clarinet | Bassoon | Horn | Oboe | Flute | Bass | Harpsichord |
|---|---|---|---|---|---|---|---|---|---|---|---|
| Notes | 83 | 51 | 51 | 51 | 41 | 36 | 41 | 28 | 37 | 43 | 51 |

Table 1: Summary statistics of the MusicNet dataset. See Sect. 2 for further discussion of MusicNet and Sect. 3 for a description of the labelling process. Appendix A discusses the methodology for computing error rate of this process.

along with an evaluation protocol. We will consider a variety of machine learning architectures for this task: i) learning from spectrogram features; ii) end-to-end learning with a neural net; iii) end-to-end learning with a convolutional neural net. Each of the proposed end-to-end models learns a set of frequency selective filters as low-level features of musical audio, which are similar in spirit to a spectrogram. The learned low-level features are visualized in Figure 1. The learned features modestly outperform spectrogram features; we will explore possible reasons for this in Sect. 5.

Figure 1: (Left) Bottom-level weights learned by a two-layer ReLU network trained on 16,384-samples windows ($\approx 1/3$ seconds) of raw audio with $\ell_2$ regularized ($\lambda = 1$) square loss for multi-label note classification on raw audio recordings. (Middle) Magnified view of the center of each set of weights. (Right) The truncated frequency spectrum of each set of weights.

## 2 MUSICNET

**Related Works.** The experiments in this paper suggest that large amounts of data are necessary to recovering useful features from music; see Sect. 4.5 for details. The Lakh dataset, released this summer based on the work of Raffel & Ellis (2015), offers note-level annotations for many 30-second clips of pop music in the Million Song Dataset (McFee et al., 2012). The syncRWC dataset is a subset of the RWC dataset (Goto et al., 2003) consisting of 61 recordings aligned to scores using the protocol described in Ewert et al. (2009). The MAPS dataset (Emiya et al., 2010) is a mixture of acoustic and synthesized data, which expressive models could overfit. The Mazurka project[3] consists of commercial music. Access to the RWC and Mazurka datasets comes at both a cost and inconvenience. Both the MAPS and Mazurka datasets are comprised entirely of piano music.

**The MusicNet Dataset.** MusicNet is a public collection of labels (exemplified in Table 2) for 330 freely-licensed classical music recordings of a variety of instruments arranged in small chamber ensembles under various studio and microphone conditions. The recordings average 6 minutes in length. The shortest recording in the dataset is 55 seconds and the longest is almost 18 minutes. Table 1 summarizes the statistics of MusicNet with breakdowns into various types of labels. Table 2 demonstrates examples of labels from the MusicNet dataset.

| Start | End | Instrument | Note | Measure | Beat | Note Value |
|-------|-------|------------|------|---------|------|------------|
| 45.29 | 45.49 | Violin | G5 | 21 | 3 | Eighth |
| 48.99 | 50.13 | Cello | A#3 | 24 | 2 | Dotted Half |
| 82.91 | 83.12 | Viola | C5 | 51 | 2.5 | Eighth |

Table 2: MusicNet labels on the Pascal String Quartet's recording of Beethoven's Opus 127, String Quartet No. 12 in E-flat major, I - Maestoso - Allegro. Creative commons use of this recording is made possible by the work of the European Archive.

MusicNet labels come from 513 label classes using the most naive definition of a class: distinct instrument/note combinations. The breakdowns reported in Table 1 indicate the number of distinct notes that appear for each instrument in our dataset. For example, while a piano has 88 keys only 83 of them are performed in MusicNet. For many tasks a note's value will be a part of its label, in which case the number of classes will expand by approximately an order of magnitude after taking the cartesian product of the set of classes with the set of values: quarter-note, eighth-note, triplet, etc. Labels regularly overlap in the time series, creating polyphonic multi-labels.

MusicNet is skewed towards Beethoven, thanks to the composer's popularity among performing ensembles. The dataset is also skewed towards Solo Piano due to an abundance of digital scores available for piano works. For training purposes, researchers may want to augment this dataset to increase coverage of instruments such as Flute and Oboe that are under-represented in MusicNet. Commercial recordings could be used for this purpose and labeled using the alignment protocol described in Sect. 3.

## 3 DATASET CONSTRUCTION

MusicNet recordings are freely-licensed classical music collected from the European Archive, the Isabella Stewart Gardner Museum, Musopen, and various artists' collections. The MusicNet labels are retrieved from digital MIDI scores, collected from various archives including the Classical Archives (`classicalarchives.com`) Suzuchan's Classic MIDI (`suzumidi.com`) and HarfeSoft (`harfesoft.de`). The methods in this section produce an alignment between a digital score and a corresponding freely-licensed recording. A recording is labeled with events in the score, associated to times in the performance via the alignment. Scores containing 6, 550, 760 additional labels are available on request to researchers who wish to augment MusicNet with commercial recordings.

Music-to-score alignment is a long-standing problem in the music research and signal processing communities (Raphael, 1999). Dynamic time warping (DTW) is a classical approach to this problem. An early use of DTW for music alignment is Orio & Schwarz (2001) where a recording is

---

[3] `http://www.mazurka.org.uk/`

aligned to a crude synthesis of its score, designed to capture some of the structure of an overtone series. The method described in this paper aligns recordings to synthesized performances of scores, using side information from a commercial synthesizer. To the best of our knowledge, commercial synthesis was first used for the purpose of alignment in Turetsky & Ellis (2003).

The majority of previous work on alignment focuses on pop music. This is more challenging than aligning classical music because commercial synthesizers do a poor job reproducing the wide variety of vocal and instrumental timbres that appear in modern pop. Furthermore, pop features inharmonic instruments such as drums for which natural metrics on frequency representations–including $\ell^2$–are not meaningful. For classical music to score alignment, a variant of the techniques described in Turetsky & Ellis (2003) works robustly. This method is described below; we discuss the evaluation of this procedure and its error rate on MusicNet in the appendix.

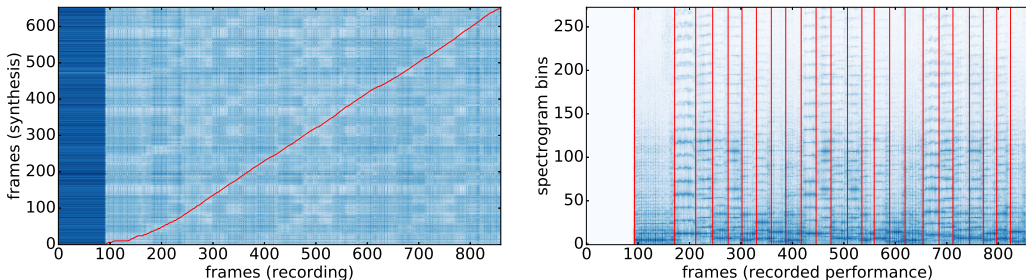

Figure 2: (Left) Heatmap visualization of local alignment costs between the synthesized and recorded spectrograms, with the optimal alignment path in red. The block from $x = 0$ to $x = 100$ frames corresponds to silence at the beginning of the recorded performance. The slope of the alignment can be interpreted as an instantaneous tempo ratio between the recorded and synthesized performances. The curvature in the alignment between $x = 100$ and $x = 175$ corresponds to an extension of the first notes by the performer. (Right) Annotation of note onsets on the spectrogram of the recorded performance, determined by the alignment shown on the left.

In order to align the performance with a score, we need to define a metric that compares short segments of the score with segments of a performance. Musical scores can be expressed as binary vectors in $E \times K$ where $E = \{1, \ldots, n\}$ and $K$ is a dictionary of notes. Performances reside in $\mathbb{R}^{T \times p}$, where $T \in \{1, \ldots, m\}$ is a sequence of time steps and $p$ is the dimensionality of the spectrogram at time $T$. Given some local cost function $C : (\mathbb{R}^p, K) \to \mathbb{R}$, a score $\mathbf{Y} \in E \times K$, and a performance $\mathbf{X} \in \mathbb{R}^{T \times p}$, the alignment problem is to

$$
\begin{aligned}
\underset{t \in \mathbb{Z}^n}{\text{minimize}} \quad & \sum_{i=1}^{n} C(\mathbf{X}_{t_i}, \mathbf{Y}_i) \\
\text{subject to} \quad & t_0 = 0, \\
& t_n = m, \\
& t_i \le t_j \qquad \text{if } i < j.
\end{aligned}
\tag{1}
$$

Dynamic time warping gives an exact solution to the problem in $\mathcal{O}(mn)$ time and space.

The success of dynamic time warping depends on the metric used to compare the score and the performance. Previous works can be broadly categorized into three groups that define an alignment cost $C$ between segments of music $\mathbf{x}$ and score $\mathbf{y}$ by injecting them into a common normed space via maps $\Psi$ and $\Phi$:

$$
C(\mathbf{x}, \mathbf{y}) = \|\Psi(\mathbf{x}) - \Phi(\mathbf{y})\|
\tag{2}
$$

The most popular approach–and the one adopted by this paper–maps the score into the space of the performance (Orio & Schwarz, 2001; Turetsky & Ellis, 2003; Soulez et al., 2003). An alternative approach maps both the score and performance into some third space, commonly a chromogram space (Hu et al., 2003; Izmirli & Dannenberg, 2010; Joder et al., 2013). Finally, some recent methods consider alignment in score space, taking $\Phi = \text{Id}$ and learning $\Psi$ (Garreau et al., 2014; Lajugie et al., 2016).

With reference to the general cost (2), we must specify the maps $\Psi, \Phi$, and the norm $\|\cdot\|$. We compute the cost in the performance feature space $\mathbb{R}^p$, hence we take $\Psi = \mathrm{Id}$. For the features, we use the log-spectrogram with a window size of 2048 samples. We use a stride of 512 samples between features. Hence adjacent feature frames are computed with 75% overlap. For audio sampled at 44.1kHz, this results in a feature representation with $44,100/512 \approx 86$ frames per second. A discussion of these parameter choices can be found in the appendix. The map $\Phi$ is computed by a synthetizer: we used Plogue's Sforzando sampler together with Garritan's Personal Orchestra 4 sample library.

For a (pseudo)-metric on $\mathbb{R}^p$, we take the $\ell^2$ norm $\|\cdot\|_2$ on the low 50 dimensions of $\mathbb{R}^p$. Recall that $\mathbb{R}^p$ represents Fourier components, so we can roughly interpret the $k$'th coordinate of $\mathbb{R}^p$ as the energy associated with the frequency $k \times (22,050/1024) \approx k \times 22.5$Hz, where $22,050$Hz is the Nyquist frequency of a signal sampled at $44.1$kHz. The 50 dimension cutoff is chosen empirically: we observe that the resulting alignments are more accurate using a small number of low-frequency bins rather than the full space $\mathbb{R}^p$. Synthesizers do not accurately reproduce the high-frequency features of a musical instrument; by ignoring the high frequencies, we align on a part of the spectrum where the synthesis is most accurate. The proposed choice of cutoff is aggressive compared to usual settings; for instance, Turetsky & Ellis (2003) propose cutoffs in the 2.5kHz range. The fundamental frequencies of many notes in MusicNet are higher than the $50 \times 22.5$Hz $\approx 1$kHz cutoff. Nevertheless, we find that all notes align well using only the low-frequency information.

## 4 METHODS

We consider identification of notes in a segment of audio $\mathbf{x} \in \mathcal{X}$ as a multi-label classification problem, modeled as follows. Assign each audio segment a binary label vector $\mathbf{y} \in \{0,1\}^{128}$. The 128 dimensions correspond to frequency codes for notes, and $\mathbf{y}_n = 1$ if note $n$ is present at the midpoint of $\mathbf{x}$. Let $f : \mathcal{X} \to \mathcal{H}$ indicate a feature map. We train a multivariate linear regression to predict $\hat{\mathbf{y}}$ given $f(\mathbf{x})$, which we optimize for square loss. The vector $\hat{\mathbf{y}}$ can be interpreted as a multi-label estimate of notes in $\mathbf{x}$ by choosing a threshold $c$ and predicting label $n$ iff $\hat{\mathbf{y}}_n > c$. We search for the value $c$ that maximizes $F_1$-score on a sampled subset of MusicNet.

### 4.1 RELATED WORK

Learning on raw audio is studied in both the music and speech communities. Supervised learning on music has been driven by access to labeled datasets. Pop music labeled with chords (Harte, 2010) has lead to a long line of work on chord recognition, most recently Korzeniowsk & Widmer (2016). Genre labels and other metadata has also attracted work on representation learning, for example Dieleman & Schrauwen (2014). There is also substantial work modeling raw audio representations of speech; a current example is Tokuda & Zen (2016). Recent work from Google DeepMind explores generative models of raw audio, applied to both speech and music (van den Oord et al., 2016).

The music community has worked extensively on a closely related problem to note prediction: fundamental frequency estimation. This is the analysis of fundamental (in contrast to overtone) frequencies in short audio segments; these frequencies are typically considered as proxies for notes. Because access to large labeled datasets was historically limited, most of these works are unsupervised. A good overview of this literature can be found in Benetos et al. (2013). Variants of non-negative matrix factorization are popular for this task; a recent example is Khlif & Sethu (2015). A different line of work models audio probabilistically, for example Berg-Kirkpatrick et al. (2014). Recent work by Kelz et al. (2016) explores supervised models, trained using the MAPS piano dataset.

### 4.2 MULTI-LAYER PERCEPTRONS

We build a two-layer network with features $f_i(\mathbf{x}) = \log\left(1 + \max(0, \mathbf{w}_i^T \mathbf{x})\right)$. We find that compression introduced by a logarithm improves performance versus a standard ReLU network (see Table 3). Figure 1 illustrates a selection of weights $w_i$ learned by the bottom layer of this network. The weights learned by the network are modulated sinusoids. This explains the effectiveness of spectrograms as a low-level representation of musical audio. The weights decay at the boundaries, analogous to Gabor filters in vision. This behavior is explained by the labeling methodology: the audio segments used here are approximately $1/3$ of a second long, and a segment is given a note

label if that note is on in the center of the segment. Therefore information at the boundaries of the segment is less useful for prediction than information nearer to the center.

## 4.3 (LOG-)SPECTROGRAMS

Spectrograms are an engineered feature representation for musical audio signals, available in popular software packages such as librosa (McFee et al., 2015). Spectrograms (resp. log-spectrograms) are closely related to a two-layer ReLU network (resp. the log-ReLU network described above). If $\mathbf{x} = (x_1, \ldots, x_t)$ denotes a segment of an audio signal of length $t$ then we can define

$$\text{Spec}_k(\mathbf{x}) \equiv \left| \sum_{s=0}^{t-1} e^{-2\pi i k s/t} x_s \right|^2 = \left( \sum_{s=0}^{t-1} \cos(2\pi k s/t) x_s \right)^2 + \left( \sum_{s=0}^{t-1} \sin(2\pi k s/t) x_s \right)^2.$$

These features are not precisely learnable by a two-layer ReLU network. But recall that $|x| = \max(0, x) + \max(0, -x)$ and if we take weight vectors $\mathbf{u}, \mathbf{v} \in \mathbb{R}^T$ with $u_s = \cos(2\pi k s/t)$ and $v_s = \sin(2\pi k s/t)$ then the ReLU network can learn

$$f_{k,\cos}(\mathbf{x}) + f_{k,\sin}(\mathbf{x}) \equiv |\mathbf{u}^T\mathbf{x}| + |\mathbf{v}^T\mathbf{x}| = \left| \sum_{s=0}^{t-1} \cos(2\pi k s/t) x_s \right| + \left| \sum_{s=0}^{t-1} \sin(2\pi k s/t) x_s \right|.$$

We call this family of features a ReLUgram and observe that it has a similar form to the spectrogram; we merely replace the $x \mapsto x^2$ non-linearity of the spectrogram with $x \mapsto |x|$. These features achieve similar performance to spectrograms on the classification task (see Table 3).

## 4.4 WINDOW SIZE

When we parameterize a network, we must choose the width of the set of weights in the bottom layer. This width is called the receptive field in the vision community; in the music community it is called the window size. Traditional frequency analyses, including spectrograms, are highly sensitive to the window size. Windows must be long enough to capture relevant information, but not so long that they lose temporal resolution; this is the classical time-frequency tradeoff. Furthermore, windowed frequency analysis is subject to boundary effects, known as spectral leakage. Classical signal processing attempts to dampen these effects with predefined window functions, which apply a mask that attenuates the signal at the boundaries (Rabiner & Schafer, 2007).

The proposed end-to-end models learn window functions. If we parameterize these models with a large window size then the model will learn that distant information is irrelevant to local prediction, so the magnitude of the learned weights will attenuate at the boundaries. We therefore focus on two window sizes: 2048 samples, which captures the local content of the signal, and 16,384 samples, which is sufficient to capture almost all relevant context (again see Figure 1).

## 4.5 REGULARIZATION

The size of MusicNet is essential to achieving the results in Figure 1. In Figure 3 (Left) we optimize a two-layer ReLU network on a small subset of MusicNet consisting of $65,000$ monophonic data points. While these features do exhibit dominant frequencies, the signal is quite noisy. Comparable noisy frequency selective features were recovered by Dieleman & Schrauwen (2014); see their Figure 3. We can recover clean features on a small dataset using heavy regularization, but this destroys classification performance; regularizing with dropout poses a similar tradeoff. By contrast, Figure 3 (Right) shows weights learned by an unregularized two-layer network trained on the full MusicNet dataset. The models described in this paper do not overfit to MusicNet and optimal performance (reported in Table 3) is achieved without regularization.

## 4.6 CONVOLUTIONAL NETWORKS

Previously, we estimated $\hat{\mathbf{y}}$ by regressing against $f(\mathbf{x})$. We now consider a convolutional model that regresses against features of a collection of shifted segments $\mathbf{x}_\ell$ near to the original segment $\mathbf{x}$. The learned features of this network are visually comparable to those learned by the fully connected network (Figure 1). The parameters of this network are the receptive field, stride, and pooling regions.

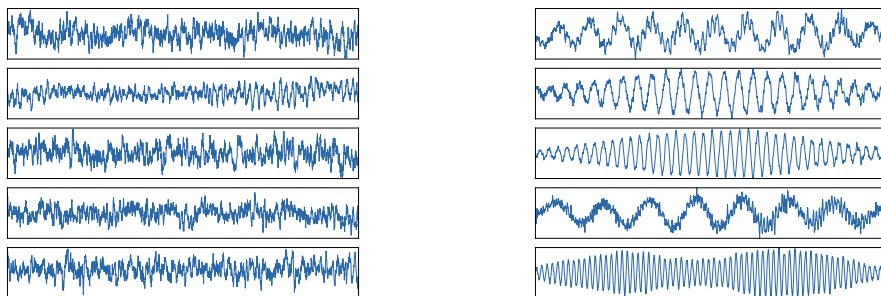

Figure 3: (Left) Features learned by a 2-layer ReLU network trained on small monophonic subset of MusicNet. (Right) Features learned by the same network, trained on the full MusicNet dataset.

The results reported in Table 3 are achieved with 500 hidden units using a receptive field of $2,048$ samples with an 8-sample stride across a window of $16,384$ samples. These features are grouped into average pools of width 16, with a stride of 8 features between pools. A max-pooling operation yields similar results. The learned features are consistent across different parameterizations. In all cases the learned features are comparable to those of a fully connected network.

## 5 RESULTS

We hold out a test set of 3 recordings for all the results reported in this section:

- Bach's Prelude in D major for Solo Piano. WTK Book 1, No 5. Performed by Kimiko Ishizaka. MusicNet recording id 2303.
- Mozart's Serenade in E-flat major. K375, Movement 4 - Menuetto. Performed by the Soni Ventorum Wind Quintet. MusicNet recording id 1819.
- Beethoven's String Quartet No. 13 in B-flat major. Opus 130, Movement 2 - Presto. Released by the European Archive. MusicNet recording id 2382.

The test set is a representative sampling of MusicNet: it covers most of the instruments in the dataset in small, medium, and large ensembles. The test data points are evenly spaced segments separated by 512 samples, between the 1st and 91st seconds of each recording. For the wider features, there is substantial overlap between adjacent segments. Each segment is labeled with the notes that are on in the middle of the segment.

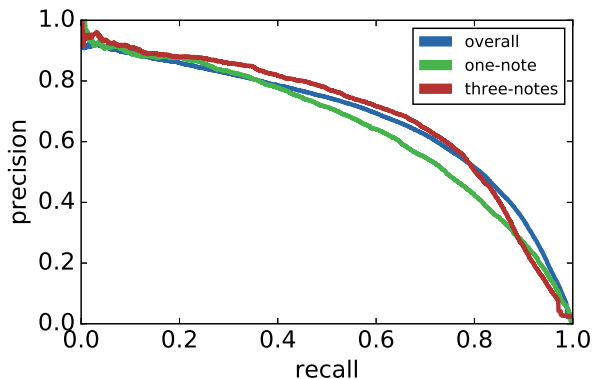

Figure 4: Precision-recall curves for the convolutional network on the test set. Curves are evaluated on subsets of the test set consisting of all data points (blue); points with exactly one label (monophonic; green); and points with exactly three labels (red).

We evaluate our models on three scores: precision, recall, and average precision. The precision score is the count of correct predictions by the model (across all data points) divided by the total number

of predictions by the model. The recall score is the count of correct predictions by the model divided by the total number of (ground truth) labels in the test set. Precision and recall are parameterized by the note prediction threshold $c$ (see Sect. 4). By varying $c$, we construct precision-recall curves (see Figure 4). The average precision score is the area under the precision-recall curve.

| Representation | Window Size | Precision | Recall | Average Precision |
|---|---|---|---|---|
| log-spectrograms | 1,024 | 49.0% | 40.5% | 39.8% |
| spectrograms | 2,048 | 28.9% | 52.5 % | 32.9% |
| log-spectrograms | 2,048 | 61.9% | 42.0% | 48.8% |
| log-ReLUgrams | 2,048 | 58.9% | 47.9% | 49.3% |
| MLP, 500 nodes | 2,048 | 50.1% | 58.0% | 52.1% |
| MLP, 2500 nodes | 2,048 | 53.6% | 62.3% | 56.2% |
| AvgPool, 2 stride | 2,148 | 53.4% | 62.5% | 56.4% |
| log-spectrograms | 8,192 | 64.2% | 28.6% | 52.1% |
| log-spectrograms | 16,384 | 58.4% | 18.1% | 45.5% |
| MLP, 500 nodes | 16,384 | 54.4% | 64.8% | 60.0% |
| CNN, 64 stride | 16,384 | 60.5% | 71.9% | 67.8% |

Table 3: Benchmark results on MusicNet for models discussed in this paper. The learned representations are optimized for square loss with SGD using the Tensorflow library (Abadi et al.). We report the precision and recall corresponding to the best $F_1$-score on validation data.

A spectrogram of length $n$ is computed from $2n$ samples, so the linear 1024-point spectrogram model is directly comparable to the MLP runs with 2048 raw samples. Learned features[4] modestly outperform spectrograms for comparable window sizes. The discussion of windowing in Sect. 4.4 partially explains this. Figure 5 suggests a second reason. Recall (Sect. 4.3) that the spectrogram features can be interpreted as the magnitude of the signal's inner product with sine waves of linearly spaced frequencies. In contrast, the proposed networks learn weights with frequencies distributed similarly to the distribution of notes in MusicNet (Figure 5). This gives the network higher resolution in the most critical frequency regions.

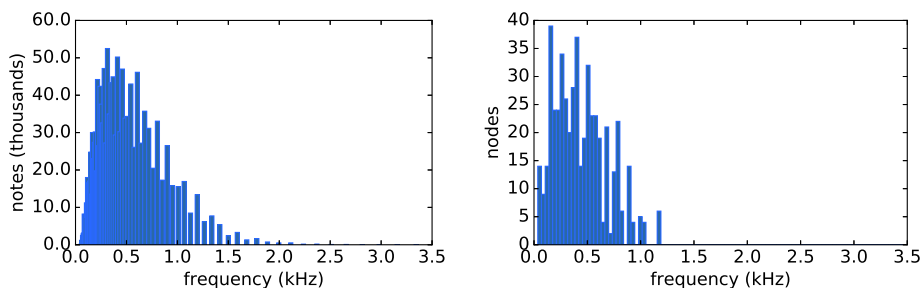

Figure 5: (Left) The frequency distribution of notes in MusicNet. (Right) The frequency distribution of learned nodes in a 500-node, two-layer ReLU network.

ACKNOWLEDGMENTS

We thank Bob L. Sturm for his detailed feedback on an earlier version of the paper. We also thank Brian McFee and Colin Raffel for fruitful discussions. Sham Kakade acknowledges funding from the Washington Research Foundation for innovation in Data-intensive Discovery. Zaid Harchaoui acknowledges funding from the program "Learning in Machines and Brains" of CIFAR.

---

[4]A demonstration using learned MLP features to synthesize a musical performance is available on the dataset webpage: `http://homes.cs.washington.edu/~thickstn/demos.html`

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

# A    VALIDATING THE MUSICNET LABELS

We validate the aligned MusicNet labels with a listening test. We create an aural representation of an aligned score-performance pair by mixing a short sine wave into the performance with the frequency indicated by the score at the time indicated by the alignment. We can listen to this mix and, if the alignment is correct, the sine tones will exactly overlay the original performance; if the alignment is incorrect, the mix will sound dissonant.

We have listened to sections of each recording in the aligned dataset: the beginning, several random samples of middle, and the end. Mixes with substantially incorrect alignments were rejected from the dataset. Failed alignments are mostly attributable to mismatches between the midi and the recording. The most common reason for rejection is musical repeats. Classical music often contains sections with indications that they be repeated a second time; in classical music performance culture, it is often acceptable to ignore these directions. If the score and performance make different choices regarding repeats, a mismatch arises. When the score omits a repeat that occurs in the performance, the alignment typically warps over the entire repeated section, with correct alignments before and after. When the score includes an extra repeat, the alignment typically compresses it into very short segment, with correct alignments on either side. We rejected alignments exhibiting either of these issues from the dataset.

From the aligned performances that we deemed sufficiently accurate to admit to the dataset, we randomly sampled 30 clips for more careful annotation and analysis. We weighted the sample to cover a wide coverage of recordings with various instruments, ensemble sizes, and durations. For each sampled performance, we randomly selected a 30 second clip. Using software transforms, it is possible to slow a recording down to approximately 1/4 speed. Two of the clips were too richly structured and fast to precisely analyze (slowing the signal down any further introduces artifacts that make the signal difficult to interpret). Even in these two rejected samples, the alignments sound substantially correct.

For the other 28 clips, we carefully analyzed the aligned performance mix and annotated every alignment error. Two of the authors are classically trained musicians: we independently checked for errors and we our analyses were nearly identical. Where there was disagreement, we used the more pessimistic author's analysis. Over our entire set of clips we averaged a $4.0\%$ error rate.

Note that we do not catch every type of error. Mistaken note onsets are more easily identified than mistaken offsets. Typically the release of one note coincides with the onset of a new note, which implicitly verifies the release. However, release times at the ends of phrases may be less accurate; these inaccuracies would not be covered by our error analysis. We were also likely to miss performance mistakes that maintain the meter of the performance, but for professional recordings such mistakes are rare.

For stringed instruments, chords consisting of more than two notes are "rolled"; i.e. they are performed serially from the lowest to the highest note. Our alignment protocol cannot separate notes that are notated simultaneously in the score; a rolled chord is labeled with a single starting time, usually the beginning of the first note in the roll. Therefore, there is some time period at the beginning of a roll where the top notes of the chord are labeled but have not yet occurred in the performance. There are reasonable interpretations of labeling under which these labels would be judged incorrect. On the other hand, if the labels are used to supervise transcription then ours is likely the desired labeling.

We can also qualitatively characterize the types of errors we observed. The most common types of errors are anticipations and delays: a single, or small sequence of labels is aligned to a slightly early or late location in the time series. Another common source of error is missing ornaments and trills: these are short flourishes in a performance are sometimes not annotated in our score data, which results in a missing annotation in the alignment. Finally, there are rare performance errors in the recordings and transcription errors in the score.

## B    ALIGNMENT PARAMETER ROBUSTNESS

The definitions of audio featurization and the alignment cost function were contingent on several parameter choices. These choices were optimized by systematic exploration of the parameter space. We investigated what happens as we vary each parameter and made the choices that gave the best results in our listening tests. Fine-tuning of the parameters yields marginal gains.

The quality of alignments improves uniformly with the quality of synthesis. The time-resolution of labels improves uniformly as the stride parameter decreases; minimization of stride is limited by system memory constraints. We find that the precise phase-invariant feature specification has little effect on alignment quality. We experimented with spectrograms and log-spectrograms using windowed and un-windowed signals. Alignment quality seemed to be largely unaffected.

The other parameters are governed by a tradeoff curve; the optimal choice is determined by balancing desirable outcomes. The Fourier window size is a classic tradeoff between time and frequency resolution. The $\ell^2$ norm can be understood as a tradeoff between the extremes of $\ell^1$ and $\ell^\infty$. The $\ell^1$ norm is too egalitarian: the preponderance of errors due to synthesis quality add up and overwhelm the signal. On the other hand, the $\ell^\infty$ norm ignores too much of the signal in the spectrogram. The spectrogram cutoff, discussed in Sec. 3, is also a tradeoff between synthesis quality and maximal use of information

## C    ADDITIONAL ERROR ANALYSIS

For each model, using the test set described in Sect. 5, we report accuracy and error scores used by the MIR community to evaluate the Multi-F0 systems. Definitions and a discussion of these metrics are presented in Poliner & Ellis (2007).

| Representation | Acc | Etot | Esub | Emiss | Efa |
|---|---|---|---|---|---|
| 512-point log-spectrogram | 28.5% | .819 | .198 | .397 | .224 |
| 1024-point log-spectrogram | 33.4% | .715 | .123 | .457 | .135 |
| 1024-point log-ReLUgram | 35.9% | .711 | .144 | .377 | .190 |
| 4096-point log-spectrogram | 24.7% | .788 | .085 | .628 | .074 |
| 8192-point log-spectrogram | 16.1% | .866 | .082 | .737 | .047 |
| MLP, 500 nodes, 2048 raw samples | 36.8% | .790 | .206 | .214 | .370 |
| MLP, 2500 nodes. 2048 samples | 40.4% | .740 | .177 | .200 | .363 |
| AvgPool, 5 stride, 2048 samples | 40.5% | .744 | .176 | .200 | .369 |
| MLP, 500 nodes, 16384 samples | 42.0% | .735 | .160 | .191 | .383 |
| CNN, 64 stride, 16384 samples | 48.9% | .634 | .117 | .164 | .352 |

Table 4: MIREX-style statistics, evaluated using the mir_eval library (Raffel et al., 2014).

# D    PRECISION & RECALL CURVES

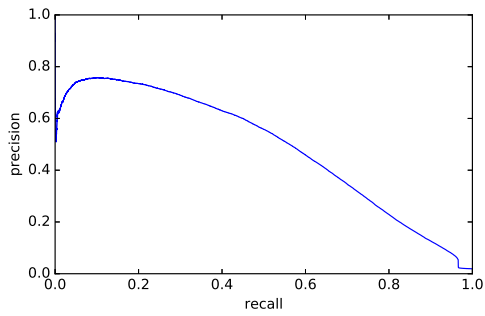

Figure 6: The linear spectrogram model.

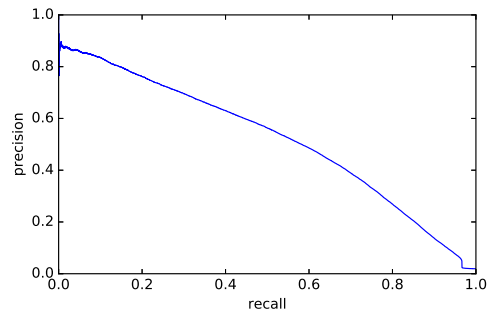

Figure 7: The 500 node, 2048 raw sample MLP.

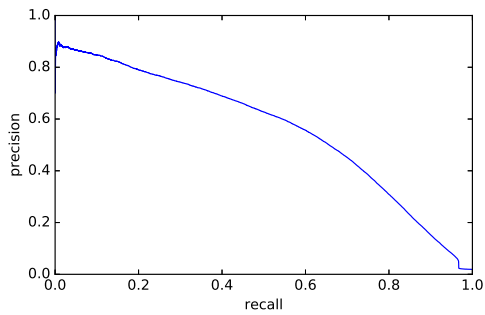

Figure 8: The 2500 node, 2048 raw sample MLP.

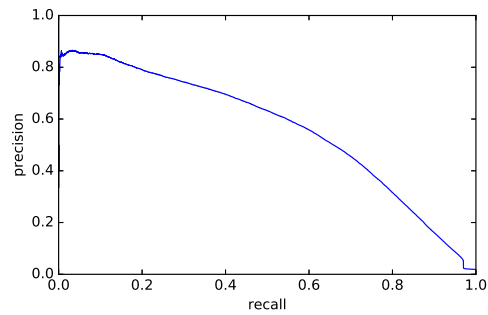

Figure 9: The average pooling model.

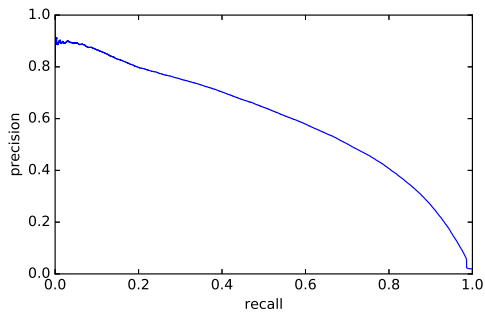

Figure 10: The 500 node, 16384 raw sample MLP.

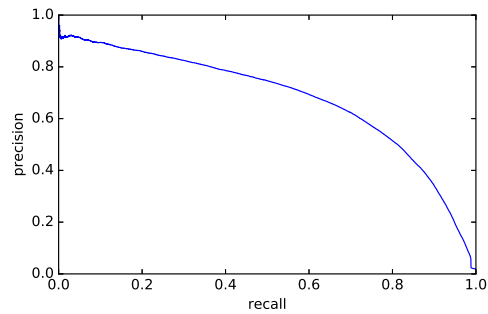

Figure 11: The convolutional model.

# E    ADDITIONAL RESULTS

We report additional results on splits of the test set described in Sect. 5.

| Model | Features | Precision | Recall | Average Precision |
|---|---|---|---|---|
| MLP, 500 nodes | 2048 raw samples | 56.1% | 62.7% | 59.2% |
| MLP, 2500 nodes | 2048 raw samples | 59.1% | 67.8% | 63.1% |
| AvgPool, 5 stride | 2048 raw samples | 59.1% | 68.2% | 64.5% |
| MLP, 500 nodes | 16384 raw samples | 60.2% | 65.2% | 65.8% |
| CNN, 64 stride | 16384 raw samples | 65.9% | 75.2% | 74.4% |

Table 5: The Soni Ventorum recording of Mozart's Wind Quintet K375 (MusicNet id 1819).

| Model | Features | Precision | Recall | Average Precision |
|---|---|---|---|---|
| MLP, 500 nodes | 2048 raw samples | 35.4% | 40.7% | 28.0% |
| MLP, 2500 nodes | 2048 raw samples | 38.3% | 44.3% | 30.9% |
| AvgPool, 5 stride | 2048 raw samples | 38.6% | 45.2% | 31.7% |
| MLP, 500 nodes | 16384 raw samples | 43.4% | 51.3% | 41.0% |
| CNN, 64 stride | 16384 raw samples | 51.0% | 57.9% | 49.3% |

Table 6: The European Archive recording of Beethoven's String Quartet No. 13 (MusicNet id 2382).

| Model | Features | Precision | Recall | Average Precision |
|---|---|---|---|---|
| MLP, 500 nodes | 2048 raw samples | 55.6% | 67.4% | 64.1% |
| MLP, 2500 nodes | 2048 raw samples | 60.1% | 71.3% | 68.6% |
| AvgPool, 5 stride | 2048 raw samples | 59.6% | 70.7% | 68.1% |
| MLP, 500 nodes | 16384 raw samples | 57.1% | 76.3% | 68.4% |
| CNN, 64 stride | 16384 raw samples | 61.9% | 80.1% | 73.9% |

Table 7: The Kimiko Ishizaka recording of Bach's Prelude in D major (MusicNet id 2303).

