# Peer review of "Learning Features of Music From Scratch"

_ICLR 2017 — accepted_

[Official Review · AnonReviewer3 · rating 6 · confidence 4 · 16 Dec 2016]
**awesome new dataset, less interesting experiments**

The paper introduces a new dataset called MusicNet (presumably analogous to ImageNet), featuring dense ground truth labels for 30+ hours of classical music, which is provided as raw audio. Such a dataset is extremely valuable for music information retrieval (MIR) research and a dataset of this size has never before been publicly available. It has the potential to dramatically increase the impact of modern machine learning techniques (e.g. deep learning) in this field, whose adoption has previously been hampered by a lack of available datasets that are large enough. The paper is clear and well-written.

The paper also features some "example" experiments using the dataset, which I am somewhat less excited about. The authors decided to focus on one single task that is not particularly challenging: identifying pitches in isolated segments of audio. Pitch information is a fairly low-level characteristic of music. Considering that isolated fragments are used as input, this is a relatively simple problem that probably doesn't even require machine learning to solve adequately, e.g. peak picking on a spectral representation could already get you pretty far. It's not clear what value the machine learning component in the proposed approach actually adds, if any. I could be wrong about this as I haven't done the comparison myself, but I think the burden is on the authors to demonstrate that using ML here is actually useful.

I would argue that one of the strenghts of the dataset is the variety of label information it provides, so a much more convincing setup would have been to demonstrate many different prediction tasks for both low-level (e.g. pitch, onsets) and high-level (e.g. composer) characteristics, perhaps with fewer and simpler models -- maybe even sticking to spectrogram input and forgoing raw audio input for the time being, as this comparison seems orthogonal to the introduction of the dataset itself. As it stands, I feel that the fact that the experiments are relatively uninteresting detracts from the main point of the paper, which is to introduce a new public dataset that is truly unique in terms of its scale and scope.

That said, the experiments seem to have been conducted in a rigorous fashion and the evaluation and analysis of the resulting models is properly executed.

Re: Section 4.5, it is rather unsurprising to me that a pitch detector would learn filters that resemble pitches (i.e. sinusoids), although the observation that this requires a relatively large amount of data is interesting. However, it would be more interesting to demonstrate that this is also the case for higher-level tasks. The authors favourably compare the features learnt by their model with prior work on end-to-end learning from raw audio, but neglect that the tasks considered in this work were much more high-level.

Some might also question whether ICLR is the appropriate venue to introduce a new dataset, but personally I think it's a great idea to submit it here, seeing as it will reach the right people. I suppose this is up to the organisers and the program committee, but I thought it important to mention this, because I don't think this paper merits acceptance based on its experimental results alone.

[Official Review · AnonReviewer1 · rating 6 · confidence 4 · 17 Dec 2016]
**Interesting corpus, what next?**

This paper describes the creation of a corpus of freely-licensed classical music recordings along with corresponding MIDI-scores aligned to the audio.  It also describes experiments in polyphonic transcription using various deep learning approaches, which show promising results.

The paper is a little disorganised and somewhat contradictory in parts. For example, I find the first sentence in section 2 (MusicNet) would better be pushed one paragraph below so that the section be allowed to begin with a survey of the tools available to researchers in music. Also, the description for Table 3 should probably appear somewhere in the Methods section. Last example: the abstract/intro says the purpose is note prediction; later (4th paragraph of intro) there's a claim that the focus is "learning low-level features of music...." I find this slightly disorienting.

Although others (Uehara et al., 2016, for example) have discussed collection platforms and corpora, this work is interesting because of its size and the approach for generating features. I'm interested in what the authors will to do expand the offerings in the corpus, both in terms of volume and diversity.

[Official Review · AnonReviewer2 · rating 8 · confidence 4 · 20 Dec 2016 (modified: 25 Jan 2017)]
**Final Review: valuable new dataset**

This paper introduces MusicNet, a new dataset. Application of ML techniques to music have been limited due to scarcity of exactly the kind of data that is provided here: meticulously annotated, carefully verified and organized, containing enough "hours" of music, and where genre has been well constrained in order to allow for sufficient homogeneity in the data to help ensure usefulness. This is great for the community.

The description of the validation of the dataset is interesting, and indicates a careful process was followed.

The authors provide just enough basic experiments to show that this dataset is big enough that good low-level features (i.e. expected sinusoidal variations) can indeed be learned in an end-to-end context.

One might argue that in terms of learning representations, the work presented here contributes more in the dataset than in the experiments or techniques used. However, given the challenges of acquiring good datasets, and given the essential role such datasets play for the community in moving research forward and providing baseline reference points, I feel that this contribution carries substantial weight in terms of expected future rewards. (If research groups were making great new datasets available on a regular basis, that would place this in a different context. But so far, that is not the case.) In otherwords, while the experiments/techniques are not necessarily in the top 50% of accepted papers (per the review criteria), I am guessing that the dataset is in the top 15% or better.

[Final Decision · Program Chairs · 06 Feb 2017]
**ICLR committee final decision**

There was some question as to weather ICLR is the right venue for this sort of dataset paper, I tend to think it would be a good addition to ICLR as people from the ICLR community are likely to be among the most interested. The problem of note identification in music is indeed challenging and the authors revised the manuscript to provide more background on the problem. No major issues with the writing or clarity. Experiments and models explored are not extremely innovative, but help create a solid dataset introduction paper.